# Overweight and obesity in south central Uganda: A population-based study

**Adeoluwa Ayoola**[1,2]*, **Robert Ssekubugu**[3,4], **Mary Kathryn Grabowski**[3,5,6],
**Joseph Ssekasanvu**[3,6], **Godfrey Kigozi**[3,4], **Aishat Mustapha**[7], **Steven J. Reynolds**[3,6,8,9],
**Anna Mia Ekstrom**[9,10,11], **Helena Nordenstedt**[10,11], **Rocio Enriquez**[10], **Ronald H. Gray**[3,6],
**Maria J. Wawer**[3,4], **Joseph Kagaayi**[3,4], **Wendy S. Post**[6,12], **Larry W. Chang**[3,6,13]

1 Johns Hopkins Bloomberg School of Public Health, Baltimore, Maryland, United States of America,
2 Stanford University School of Medicine, Stanford, California, United States of America, 3 Rakai Health
Sciences Program, Kalisizo, Uganda, 4 Department of Epidemiology and Biostatistics, School of Public
Health, Makerere University, Kampala, Uganda, 5 Department of Pathology, Johns Hopkins School of
Medicine, Baltimore, Maryland, United States of America, 6 Department of Epidemiology, Johns Hopkins
Bloomberg School of Public Health, Baltimore, Maryland, United States of America, 7 Department of Internal
Medicine, Johns Hopkins School of Medicine, Baltimore, Maryland, United States of America, 8 Division of
Intramural Research, National Institute for Allergy and Infectious Diseases, National Institutes of Health,
Bethesda, Maryland, United States of America, 9 Department of Infectious Diseases, South Central Hospital,
Stockholm, Sweden, 10 Department of Global Public Health, Karolinska Institute, Stockholm, Sweden,
11 Department of Medicine and Infectious Diseases, Danderyd University Hospital, Stockholm, Sweden,
12 Division of Cardiology, Department of Medicine, Johns Hopkins School of Medicine, Baltimore, Maryland,
United States of America, 13 Division of Infectious Diseases, Department of Medicine, Johns Hopkins School
of Medicine, Baltimore, Maryland, United States of America

* adeoluwa@stanford.edu, aayoola3@jh.edu

pgph.0001051

MALAYSIA

**Data Availability Statement:** A deidentified version
of the data may be provided to interested parties
subject through completion of the Rakai Health
Sciences Program data request form and signing

## Abstract

Obesity is a rapidly growing global health challenge, but there are few population-level studies from non-urban settings in sub-Saharan Africa. We evaluated the prevalence of overweight (body mass index (BMI)>25 kg/m$^2$), obesity (BMI>30 kg/m$^2$), and associated factors using data from May 2018 to November 2020 from the Rakai Community Cohort Study, a population-based cohort of residents aged 15 to 49 living in forty-one fishing, trading, and agrarian communities in South Central Uganda. Modified Poisson regression was used to estimate adjusted prevalence risk ratios (PRR) and 95% confidence intervals (CI) in 18,079 participants. The overall mean BMI was 22.9 kg/m$^2$. Mean BMI was 21.5 kg/m$^2$ and 24.1 kg/m$^2$ for males and females, respectively. The prevalence of overweight and obesity were 22.8% and 6.2%, respectively. Females had a higher probability of overweight/obesity (PRR: 4.11, CI: 2.98–5.68) than males. For female participants, increasing age, higher socioeconomic status, residing in a trading or fishing community (PRR: 1.25, CI 1.16–1.35 and PRR: 1.17, CI 1.10–1.25, respectively), being currently or previously married (PRR: 1.22, CI 1.07–1.40 and PRR: 1.16, CI 1.01–1.34, respectively), working in a bar/restaurant (PRR: 1.29, CI 1.17–1.45), trading/shopkeeping (PRR: 1.38, CI 1.29–1.48), and reporting alcohol use in the last year (PRR: 1.21, CI 1.10–1.33) were risk factors for overweight/obese. For male participants, increasing age, higher socioeconomic status, being currently married (PRR: 1.94, CI 1.50–2.50), residing in a fishing community (PRR: 1.68, CI 1.40–2.02), working in a bar/restaurant (PRR: 2.20, CI 1.10–4.40), trading/shopkeeping (PRR: 1.75, CI 1.45–2.11), or fishing (PRR: 1.32, CI 1.03–1.69) increased the probability of

of a Data Transfer Agreement. Inquiries should be directed to datarequests@rhsp.org.

**Funding:** This work was supported by the National Heart, Lung, and Blood Institute (grant number R01HL107275, recipient LC), the Division of Intramural Research, National Institute of Allergy and Infectious Diseases (recipient SR), National Institutes of Health, the Swedish Research Council (grant numbers 2015-05864, 2016-05647, recipient HN), and the Johns Hopkins University Center for AIDS Research (P30AI094189). The funders had no role in study design, data collection and analysis, decision to publish, or preparation of the manuscript.

**Competing interests:** The authors declare that they have no competing interests.

overweight/obesity. Non-Muslim participants, male smokers, and HIV-positive females had a lower probability of overweight/obese. The prevalence of overweight/obesity in non-urban Ugandans is substantial. Targeted interventions to high-risk subgroups in this population are needed.

## Introduction

Obesity is a rapidly growing global health challenge with grave health consequences [1–3]. Potential health complications include hypertension, diabetes, cardiovascular disease, and major cancers [4]. In Uganda, overweight and obesity rates have increased [5, 6], with one study showing a doubling of overweight/obesity rates between 1995 and 2011, from 8% to 18% [7, 8].

The growing prevalence of overweight and obesity in low- and middle- income countries has been associated with increased urbanization, access to high-caloric diets, and lower physical activity [3, 9, 10]. Predictors of risk of obesity in Sub-Saharan Africa from past literature include female sex [5, 6, 11, 12], living in urban areas [3, 5, 7], older age [5], higher education [3], higher income [3, 7], and being married [3]. In rural communities of Eastern Uganda, similar factors were associated with overweight/obesity [13]. One Ugandan study found a higher prevalence of abdominal fat in women, married or cohabiting people, and urban dwellers [7].

Most prior studies of overweight and obesity had small sample sizes, were not population-based, and did not assess certain sociodemographic factors needed for designing appropriate public health interventions, especially in rural communities, as the prevalence of cardiovascular disease increases in sub-Saharan Africa [14]. This study aimed to determine the prevalence of overweight and obesity in South Central Uganda and associated sociodemographic factors for being overweight or obese to facilitate policies and programs to address this issue in Uganda.

## Materials and methods

### Sample inclusion and criteria

The study used cross-sectional data from the Rakai Community Cohort Study (RCCS), collected from May 2018 to November 2020. RCCS is an open population-based, longitudinal cohort study that began initially in 1994 to study and address the growing HIV/AIDS epidemic in Rakai, Uganda. A full description of the RCCS design and data collection procedures has been described in previous literature [15, 16]. Briefly, RCCS holds informational community mobilization events and periodic censuses in the Rakai district, during which all households from the study communities are approached by the study team for recruitment. Eligible cohort participants are identified and enrolled. RCCS collects key health data of adult residents (aged 15 to 49 years) who provide written informed consent, using periodic interviews to assess demographics, sexual and health-seeking behaviors, and uptake of HIV prevention and treatment services. Pregnant women (992) and participants with incomplete data (62) were excluded from this analysis.

### Calculation of body mass index

Height was measured in centimeters on a flat surface, with participant shoes removed, using a stadiometer. Weight was measured in kilograms using a portable weighing scale (SECA scale,

model 762 1019008) on a flat surface, with participants in light clothing and without shoes. BMI was calculated as weight divided by height squared (kg/m$^2$) using the World Health Organization (WHO) defined standards of underweight (BMI <18.5 kg/m$^2$), normal weight (BMI 18.5 kg/m$^2$–24.9 kg/m$^2$), overweight (BMI 25.0 kg/m$^2$–29.9 kg/m$^2$) and obese (BMI $\geq$30.0 kg/m$^2$), respectively [17].

## Covariates

Community type included agrarian, trading, and fishing communities [15]. Occupations for individual participants were categorized into the five groups according to the most common occupations: agriculture/housework, bar and restaurant work, fishing, trading/shopkeeping, and other (i.e., government/clerical/teaching, student, medical worker, military/police, hairdresser/salon owners, mechanics). Education level was categorized as none, primary, and secondary or above. Socioeconomic status (SES) was calculated using a household asset-based measure [18]. Marital status was categorized into "Never married," "Currently married," and "Previously married" (i.e., separated, divorced, or widowed). Past pregnancy was assessed with the question "Have you ever been pregnant (including current pregnancy)?". HIV serostatus was determined using a rapid antigen test algorithm with confirmatory immunoassay testing [19]. Current antiretroviral therapy (ART) use was assessed using a list of ART medications.

Current smoking status (i.e., cigarettes, tobacco, or pipe smoking) was assessed, and alcohol use was determined in the past year, stratified by recency of use. Participants self-reported religion as Muslim or non-Muslim (i.e., None, Catholic, Protestant, Saved/Pentecostal, or Other).

## Analysis

Data analysis was performed using STATA/BE (version 17.0 2021, StataCorp LLC, College Station, Texas). The prevalence of overweight/obesity and obesity was calculated as the percentage of individuals with BMI $\geq$25 kg/m$^2$ and BMI $\geq$30kg/m$^2$, respectively. Regression analysis was performed separately for males and females. The prevalence of overweight and/or obesity was the primary outcome. Secondary outcomes included the prevalence of underweight and prevalence of obesity alone. Covariates, selected based on prior literature, included age, community type, SES, occupation, education level, marital status, religion, current smoking, HIV status, current ART, current and last alcohol use. Past pregnancy was a covariate for females.

The prevalence of overweight, obesity, and underweight were calculated as a percentage of the study population. To evaluate factors associated with underweight, overweight and obesity, we used a modified Poisson model with a log link function to estimate prevalence risk ratios (PRR) [20]. To estimate the association between sex and obesity, a combined model with both males and females was also fitted. This combined model estimated a coefficient for the sex and included interactions for significant covariates that differed between the final male and female stratified models, controlling for other covariates (see S1 File). Regression models estimated unadjusted PRRs and 95% confidence intervals (CIs). Covariates with p-value <0.05 were included in the adjusted models. Multicollinearity in the final adjusted models was tested using the variable inflation factor (VIF). All VIFs were below 5, indicating acceptably low levels of collinearity. P-values <0.05 were considered statistically significant.

## Ethics statement

This study was approved by the Research and Ethics Committee of the Uganda Virus Research Institute (UVRI: GC/127/08/12/137), the Ugandan National Council for Science and Technology (UNCST HS 540), and the Johns Hopkins University School of Medicine Institutional Review Board (IRB00217467). Written informed consent was obtained from all adult

participants and emancipated participants less than 18 years of age. For unemancipated participants less than 18 years of age, written informed consent was obtained from their parent/guardian and written informed assent was obtained from them.

## Results

### Participant characteristics

Table 1 presents the characteristics of the 18,062 participants included in the study. Fifty-one percent were female, the mean age was 30.1 years (standard deviation (SD): 0.07 years, range 15–49). About a third (31%) of participants were ages 20–29 years, while 21.5% were age 40–49 years old. Forty-six percent were from agrarian, 33% from trading, and 21% percent from fishing communities. The largest proportion of participants came from low SES households (31.8%), were agricultural or houseworkers (38.5%), currently married (53.5%), had a primary education (58.0%), and identified as non-Muslim (86.3%). Most participants were never smokers (92.5%) and HIV seronegative (82.6%). Most had not used alcohol in over a year (56.1%) and most of the women reported a past pregnancy (80.3%).

### Prevalence of overweight and obesity

The overall mean BMI was 22.8 kg/m$^2$ (SD 4.1 kg/m$^2$). The mean for males was 21.5 kg/m$^2$ (SD 2.9 kg/m$^2$), while the mean for females was 24.1 kg/m$^2$ (SD 4.6 kg/m$^2$). Overall, 9% of study participants were underweight, 68% were normal weight, approximately 17% were overweight, and 6% were obese (Table 1). Therefore, 23% of all participants were overweight or obese. Tables 2 and 3 present counts and percentage distributions of participants in each BMI category by sociodemographic characteristics. Approximately 24% of females were overweight and 11% were obese while approximately 9% and 1% of males were overweight and obese, respectively (Tables 2 and 3). The combined prevalence of overweight and obesity increased from 14% in 15–19-year-old females to 42% and 47% in females aged 30–39 and 40–49 years, respectively. For males, the prevalence of overweight and obesity increased from 2% in 15–19-year-old males to 15% and 14% in males aged 30–39 and 40–49 years, respectively.

### Factors associated with the probability of being overweight/obese and obese

All results are conditional on the other covariates in the model. Females had a higher probability of being overweight/obese (classified as BMI$\geq$ 25 kg/m$^2$) or obese, compared to males. Specifically, males had a 76% lower probability of being overweight/obese (adjusted PRR: 0.24, CI: 0.18, 0.34) and 86% lower probability of being obese (adjusted PRR: 0.14, CI: 0.18, 0.34) compared to females (Table 4). Males had a 75% higher probability of being underweight (adjusted PRR: 1.75; CI: 1.56, 1.96) compared to females (Table 4).

   Tables 5–7 show the unadjusted and adjusted PRRs by participant characteristic for being overweight/obese, obese, and underweight. The probability of being overweight/obese or obese was higher with age for both males and females. Males in the 40–49 years age group were 4.91 times more likely to be overweight/obese compared to males in the 15–19 years age group (CI: 3.14, 7.68). Currently married individuals had a higher probability of being overweight/obese compared to never married individuals. Specifically, currently married males had a 1.94 times higher probability of being overweight/obese compared to never married males (CI: 1.50, 2.50). Currently married females also had a higher probability of being overweight/obese, as did previously married females. Female participants who were currently or previously married had a higher probability of obesity, but not male participants.

**Table 1. Participants' demographics and characteristics.**

| CHARACTERISTIC | COUNT (PERCENT) |
|---|---|
| **SEX** | |
| MALE | 8,767 (48.5) |
| FEMALE | 9,295 (51.5) |
| **AGE GROUP, YEARS** | |
| 15–19 | 3,410 (18.9) |
| 20–29 | 5,535 (30.6) |
| 30–39 | 5,239 (29.0) |
| 40–49 | 3,878 (21.5) |
| MEAN +/- SD | 30.12 +/- 0.073 |
| **COMMUNITY TYPE** | |
| AGRARIAN | 8,314 (46.0) |
| FISHING | 3,826 (21.2) |
| TRADING | 5,922 (32.8) |
| **SOCIOECONOMIC STATUS** | |
| LOWEST | 5,744 (31.8) |
| LOW-MIDDLE | 4,236 (23.4) |
| HIGH-MIDDLE | 3,876 (21.5) |
| HIGHEST | 4,214 (23.3) |
| **OCCUPATION** | |
| AGRIC\HOUSEWORK | 6,945 (38.5) |
| BAR\RESTAURANT | 664 (3.7) |
| FISHING | 1,266 (7.0) |
| TRADE\SHOPKEEPER | 2,786 (15.4) |
| OTHER | 6,401 (35.4) |
| **MARITAL STATUS** | |
| NEVER | 5,255 (29.1) |
| CURRENTLY MARRIED | 9,668 (53.5) |
| PREVIOUSLY MARRIED | 3,139 (17.4) |
| **EDUCATION** | |
| NONE | 1,256 (6.95) |
| PRIMARY | 10,479 (58.0) |
| SECONDARY AND ABOVE | 6,327 (35.0) |
| **RELIGION** | |
| MUSLIM | 2,469 (13.7) |
| NON-MUSLIM | 15,593 (86.3) |
| **PAST PREGNANCY** | |
| YES | 7,463 (80.3) |
| NO | 1,832 (19.7) |
| TOTAL | 9,295 |
| **CURRENT SMOKER** | |
| NO | 16,704 (92.5) |
| YES | 1,358 (7.5) |
| **HIV SEROSTATUS** | |
| NEGATIVE | 14,911 (82.6) |
| POSITIVE | 3,151 (17.4) |
| **CURRENT ART USE** | |
| YES | 2,654 (84.2) |

(*Continued*)

**Table 1.** (Continued)

| CHARACTERISTIC | COUNT (PERCENT) |
|---|---|
| NO | 497 (15.8) |
| TOTAL | 3,151 |
| **DRINKS ALCOHOL** | |
| NO | 10,125 (56.1) |
| YES | 7,937 (43.9) |
| **LAST ALCOHOL USE** | |
| MORE THAN 12 MONTHS | 10,125 (56.1) |
| WITHIN THE LAST 12 MONTHS | 1,273 (7.0) |
| 1–4 WEEKS | 1,751 (9.7) |
| 0 DAYS–1 WEEK | 4,913 (27.2) |
| **BMI** | |
| UNDERWEIGHT | 1,625 (9.0) |
| NORMAL WEIGHT | 12,329 (68.3) |
| OVERWEIGHT | 2,984 (16.5) |
| OBESE | 1,123 (6.2) |
| **TOTAL** | 18,062 (100.0) |

Participants sociodemographic data (N = 18,709)

Males in fishing communities had a 1.68 times higher probability of being overweight/obese compared to males in agrarian communities (CI: 1.40, 2.02). Females residing in fishing and trading communities had a higher probability of being overweight/obese and obese compared to those residing in agrarian communities (adjusted PRR 1.25 and 1.17, respectively). As SES increased, the probability of being overweight/obese and obese became higher. Compared to the lowest SES, males in the highest SES had 2.52 times higher probability of being overweight/obese (CI: 2.06, 3.07). Although SES presented a higher probability of obesity for high-middle or high SES males, there was no difference in the probability of obesity alone for males in the low-middle SES group compared to those in low SES group (adjusted PRR: 1.62; CI: 0.78, 3.34). Females also had a higher probability of being overweight/obese and obese with increasing SES. Females in the highest SES group were 1.72 times more likely to be overweight/obese compared to females in the lowest SES group (CI: 1.59, 1.87). Females who worked in bar/restaurant jobs or in trading/shopkeeping had a higher probability of overweight/obesity and obesity. For males, occupations in fishing and trading/shopkeeping (but not bar/restaurant work) were associated with a higher probability of being overweight/obese, but not a higher probability of being obese alone. Females who reported alcohol intake in the past year had a higher probability of being overweight/obese. Additionally, females who reported alcohol use in the last week had a higher probability of obesity.

Male smokers had a lower probability of being overweight/obese, as did females who identified as non-Muslim. Females living with HIV had a lower probability of obesity compared to HIV negative participants. For females, having a past pregnancy was not associated with being overweight or obese. For all participants, current ART use had no significant associations with overweight or obesity.

### Factors associated with the probability of being underweight

Males had a higher prevalence of underweight, BMI $<18.5$ kg/m$^2$, (11.7%) compared to females (6.5%). Smoking was associated with a higher probability of being underweight for

**Table 2. Female population by BMI category over sociodemographic characteristics, including mean BMI.**

| FEMALE PARTICIPANT CHARACTERISTICS | UNDERWEIGHT (%) | NORMAL (%) | OVERWEIGHT (%) | OBESE (%) | OVERWEIGHT/OBESE (%) | MEAN BMI | TOTAL |
|---|---|---|---|---|---|---|---|
| **AGE GROUP, YEARS** | | | | | | | |
| 15–19 | 194 (11.5%) | 1,260 (74.9%) | 201 (11.9%) | 28 (1.7%) | 229 (13.6%) | 21.86 | 1,683 |
| 20–29 | 168 (5.9%) | 1,756 (61.9%) | 671 (23.7%) | 241 (8.5%) | 912 (32.2%) | 23.72 | 2,836 |
| 30–39 | 113 (4.1%) | 1,430 (51.6%) | 781 (28.2%) | 448 (16.2%) | 1,229 (44.4%) | 25.15 | 2,772 |
| 40–49 | 127 (6.3%) | 1,007 (50.2%) | 561 (28.0%) | 309 (15.4%) | 870 (43.4%) | 24.95 | 2,004 |
| **COMMUNITY TYPE** | | | | | | | |
| AGRARIAN | 333 (7.8%) | 2,659 (62.0%) | 918 (21.4%) | 382 (8.9%) | 1,300 (29.3%) | 23.66 | 4,292 |
| FISHING | 75 (4.3%) | 943 (54.6%) | 475 (27.5%) | 234 (13.5%) | 709 (41.0%) | 24.67 | 1,727 |
| TRADING | 194 (5.9%) | 1,851 (56.5%) | 821 (25.1%) | 410 (12.5%) | 1,231 (37.6%) | 24.30 | 3,276 |
| **SOCIOECONOMIC STATUS** | | | | | | | |
| LOWEST | 243 (8.6%) | 1,834 (65.2%) | 554 (19.7%) | 183 (6.5%) | 737 (26.2%) | 23.10 | 2,814 |
| LOW-MIDDLE | 135 (6.0%) | 1,337 (59.6%) | 551 (24.5%) | 222 (9.9%) | 773 (34.4%) | 23.95 | 2,245 |
| HIGH-MIDDLE | 101 (4.9%) | 1,118 (54.1%) | 538 (26.1%) | 308 (14.9%) | 846 (41.0%) | 24.84 | 2,065 |
| HIGHEST | 123 (5.7%) | 1,164 (53.6%) | 571 (26.3%) | 313 (14.4%) | 884 (40.7%) | 24.73 | 2,171 |
| **OCCUPATION** | | | | | | | |
| AGRIC/HOUSEWORK | 295 (6.8%) | 2,660 (61.1%) | 1,003 (23.0%) | 399 (9.2%) | 1,402 (32.2%) | 23.78 | 4,357 |
| BAR/RESTAURANT | 33 (5.1%) | 309 (48.3%) | 189 (29.5%) | 109 (17.0%) | 298 (46.5%) | 25.28 | 640 |
| TRADE/SHOPKEEPER | 49 (3.0%) | 775 (47.4%) | 489 (29.9%) | 321 (19.6%) | 810 (49.5%) | 25.72 | 1,634 |
| OTHER | 225 (8.4%) | 1,709 (64.2%) | 533 (20.0%) | 197 (7.4%) | 730 (27.4%) | 23.24 | 2,664 |
| **MARITAL STATUS** | | | | | | | |
| NEVER MARRIED | 251 (11.6%) | 1,516 (70.2%) | 320 (14.8%) | 72 (3.3%) | 392 (18.1%) | 22.24 | 2,159 |
| PREVIOUSLY MARRIED | 114 (5.9%) | 1,064 (54.8%) | 510 (26.2%) | 254 (13.1%) | 764 (39.3%) | 24.48 | 1,943 |
| CURRENTLY MARRIED | 237 (4.6%) | 2,872 (55.3%) | 1,384 (26.7%) | 700 (13.5%) | 2,084 (40.2%) | 24.69 | 5,193 |
| **EDUCATION** | | | | | | | |
| NONE | 46 (6.4%) | 427 (59.0%) | 166 (22.9%) | 85 (11.7%) | 251 (34.6%) | 24.03 | 724 |
| PRIMARY | 352 (6.9%) | 2,988 (58.9%) | 1,202 (23.7%) | 529 (10.4%) | 1,731 (34.1%) | 24.00 | 5,071 |
| SECONDARY AND ABOVE | 204 (5.8%) | 2,038 (58.2%) | 846 (24.2%) | 412 (11.8%) | 1,258 (36.0%) | 24.19 | 3,500 |
| **RELIGION** | | | | | | | |
| MUSLIM | 71 (5.7%) | 712 (56.7%) | 311 (24.8%) | 161 (12.8%) | 472 (37.6%) | 24.46 | 1,255 |
| NON-MUSLIM | 531 (6.6%) | 4,741 (59.0%) | 1,903 (23.7%) | 865 (10.8%) | 2,768 (34.5%) | 24.01 | 8,040 |
| **PAST PREGNANCY** | | | | | | | |
| NO | 216 (11.8%) | 1,312 (71.6%) | 250 (13.6%) | 54 (2.9%) | 304 (16.5%) | 22.10 | 1,832 |
| YES | 386 (5.2%) | 4,141 (55.5%) | 1,964 (26.3%) | 972 (13.0%) | 2,936 (39.3%) | 24.56 | 7,463 |
| **CURRENT SMOKER** | | | | | | | |
| NO | 582 (6.4%) | 5,305 (58.7%) | 2,154 (23.8%) | 997 (11.0%) | 3,151 (34.8%) | 24.08 | 9,038 |
| YES | 20 (7.8%) | 148 (57.6%) | 60 (23.3%) | 29 (11.3%) | 89 (34.6%) | 23.98 | 257 |
| **HIV SEROSTATUS** | | | | | | | |
| NEGATIVE | 461 (6.3%) | 4,303 (58.7%) | 1,729 (23.6%) | 839 (11.4%) | 2,568 (35.0%) | 24.13 | 7,332 |
| POSITIVE | 141 (7.2%) | 1,150 (58.6%) | 485 (24.7%) | 187 (9.5%) | 672 (34.2%) | 23.86 | 1,963 |
| **CURRENT ART USE** | | | | | | | |
| NO | 20 (7.4%) | 156 (58.0%) | 71 (26.4%) | 22 (8.2%) | 93 (34.6%) | 23.83 | 269 |
| YES | 121 (7.1%) | 994 (58.7%) | 414 (24.4%) | 165 (9.7%) | 579 (34.1%) | 23.86 | 1,694 |
| **DRINKS ALCOHOL** | | | | | | | |
| NO | 444 (7.3%) | 3,774 (62.2%) | 1,298 (21.4%) | 548 (9.0%) | 1,846 (30.4%) | 23.63 | 6,064 |
| YES | 158 (4.9%) | 1,679 (52.0%) | 916 (28.4%) | 478 (14.8%) | 1,394 (43.1%) | 24.91 | 3,231 |
| **LAST ALCOHOL USE** | | | | | | | |
| 0 DAYS–1 WEEK | 70 (4.4%) | 791 (49.7%) | 451 (28.3%) | 281 (17.6%) | 732 (45.9%) | 25.33 | 1,593 |

*(Continued)*

**Table 2.** (Continued)

| FEMALE PARTICIPANT CHARACTERISTICS | UNDERWEIGHT (%) | NORMAL (%) | OVERWEIGHT (%) | OBESE (%) | OVERWEIGHT/OBESE (%) | MEAN BMI | TOTAL |
|---|---|---|---|---|---|---|---|
| 1–4 WEEKS | 53 (5.7%) | 503 (54.2%) | 255 (27.5%) | 117 (12.6%) | 372 (40.2%) | 24.49 | 928 |
| WITHIN THE LAST 12 MONTHS | 35 (4.9%) | 385 (54.2%) | 210 (29.6%) | 80 (11.3%) | 290 (41.9%) | 24.50 | 710 |
| MORE THAN 12 MONTHS | 444 (7.3%) | 3,774 (62.2%) | 1,298 (21.4%) | 548 (9.0%) | 1,846 (30.4%) | 23.63 | 6,064 |
| TOTAL | 602 (6.5%) | 5453 (58.8%) | 2214 (23.8%) | 1026 (11.0%) | 3,240 (34.8%) | 24.07 | 9,295 |

Female BMI groups by demographic characteristics. HIV = Human Immunodeficiency Virus. ART = anti-retroviral therapy. Note that row percentages may not total to 100% due to rounding.

male participants, with smokers being 1.91 times more likely to be underweight than non-smokers (CI = 1.60, 2.27). Being from a fishing community, a higher SES group, currently married, or having trade/shopkeeping occupation were associated with lower probability of being underweight for both males and females. For females only, being from a trading community or being previously married were associated with lower probability of underweight. HIV status was not associated with being underweight for males or females.

## Discussion

Preventing and reducing the prevalence of overweight and obesity is an important public health issue. In this study, one of the largest to date in sub-Saharan Africa, the mean BMI was 22.9 kg/m$^2$ and over 68% of the population fell within the normal BMI category. However, we also observed substantial prevalence of overweight/obesity in this non-urban population-based cohort, comparable or higher than previous studies of overweight and obesity in rural and peri-urban communities in Uganda [13, 21].

Older age was consistently associated with a greater prevalence of overweight and obesity, consistent with past studies in Uganda [13, 21] and other sub-Saharan African countries [22–25]. Female sex was also associated with higher probability of overweight and obesity. Past literature hypothesized reasons for the sex disparity, including hormonal differences between men and women leading to increased fat accumulation for women, lower levels of physical activity [9, 26], and cultural norms, particularly in African countries [9, 27, 28]. In Uganda, and many communities in sub-Saharan Africa, a married woman's weight may be seen as a reflection of her family's wealth and status [29], and a larger body stature may be desired [30, 31]. For women, past pregnancy was not associated with a higher probability of overweight/obesity in the adjusted model, but being multiparous, although not captured by this study, may increase the probability of obesity [32].

Being currently married was found to be significantly associated with a greater probability of being overweight or obese compared with never married individuals, consistent with prior African studies [7, 25, 29]. One explanation for this is the social obligations hypothesis, which proposes that the social stability and spousal obligations to share meals may contribute to increased BMI for married individuals [33]. Further, males who were separated, divorced, or widowed did not experience a higher probability of overweight and/or obesity with marriage, which is consistent with the marriage market hypothesis that return to the "marriage market" may promote weight loss or a healthy BMI to attract a more favorable spouse, especially among men [33, 34].

Muslim religion was associated with higher probability of being overweight and/or obese compared to non-Muslims. Religiosity has been previously linked to higher BMI [23, 35].

**Table 3. Male population by BMI category over sociodemographic characteristics, including mean BMI.**

| MALE PARTICIPANT CHARACTERISTICS | UNDERWEIGHT (%) | NORMAL (%) | OVERWEIGHT (%) | OBESE (%) | OVERWEIGHT/OBESE (%) | MEAN BMI | TOTAL |
|---|---|---|---|---|---|---|---|
| **AGE GROUP, YEARS** | | | | | | | |
| 15–19 | 409 (23.7%) | 1,285 (74.4%) | 31 (1.8%) | 2 (0.1%) | 33 (1.9%) | 20.06 | 1,727 |
| 20–29 | 181 (6.7%) | 2,305 (85.4%) | 201 (7.4%) | 12 (0.4%) | 213 (7.8%) | 21.67 | 2,699 |
| 30–39 | 196 (7.9%) | 1,911 (77.5%) | 320 (13.0%) | 40 (1.6%) | 360 (14.6%) | 22.12 | 2,467 |
| 40–49 | 237 (12.6%) | 1,375 (73.4%) | 219 (11.7%) | 43 (2.3%) | 262 (14.0%) | 21.84 | 1,874 |
| **COMMUNITY TYPE** | | | | | | | |
| AGRARIAN | 558 (13.9%) | 3,132 (77.9%) | 295 (7.3%) | 37 (0.9%) | 332 (8.2%) | 21.26 | 4,022 |
| FISHING | 127 (6.1%) | 1,700 (81.0%) | 244 (11.6%) | 28 (1.3%) | 272 (12.9%) | 22.11 | 2,099 |
| TRADING | 338 (12.7%) | 2,044 (77.2%) | 232 (8.8%) | 32 (1.2%) | 264 (10.0%) | 21.43 | 2,646 |
| **SOCIOECONOMIC STATUS** | | | | | | | |
| LOWEST | 403 (13.9%) | 2,364 (80.7%) | 149 (5.1%) | 14 (0.5%) | 163 (5.6%) | 21.07 | 2,930 |
| LOW-MIDDLE | 246 (12.4%) | 1,555 (78.1%) | 174 (8.7%) | 16 (0.8%) | 190 (8.5%) | 21.43 | 1,991 |
| HIGH-MIDDLE | 155 (8.6%) | 1,397 (77.1%) | 227 (12.5%) | 32 (1.8%) | 259 (14.3%) | 22.01 | 1,811 |
| HIGHEST | 219 (10.8%) | 1,560 (76.7%) | 221 (10.9%) | 35 (1.7%) | 256 (12.6%) | 21.80 | 2,035 |
| **OCCUPATION** | | | | | | | |
| AGRIC/HOUSEWORK | 327 (12.6%) | 2,070 (80.0%) | 174 (6.7%) | 17 (0.7%) | 191 (7.4%) | 21.24 | 2,588 |
| BAR/RESTAURANT | 4 (16.7%) | 15 (62.5%) | 4 (16.7%) | 1 (4.2%) | 5 (20.9%) | 21.90 | 24 |
| FISHING | 65 (5.1%) | 1,061 (83.8%) | 129 (10.2%) | 11 (0.9%) | 140 (11.1%) | 22.05 | 1,266 |
| TRADE/SHOPKEEPER | 72 (6.2%) | 878 (76.2%) | 176 (15.3%) | 26 (2.3%) | 202 (17.6%) | 22.49 | 1,152 |
| OTHER | 555 (14.9%) | 2,852 (76.3%) | 288 (7.7%) | 42 (1.1%) | 330 (8.8%) | 21.23 | 3,737 |
| **MARITAL STATUS** | | | | | | | |
| NEVER MARRIED | 525 (17.0%) | 2,460 (79.5%) | 103 (3.3%) | 8 (0.3%) | 111 (3.6%) | 20.66 | 3,096 |
| PREVIOUSLY MARRIED | 142 (11.9%) | 962 (80.4%) | 83 (6.9%) | 9 (0.8%) | 92 (7.7%) | 21.32 | 1,196 |
| CURRENTLY MARRIED | 356 (8.0%) | 3,454 (77.2%) | 585 (13.1%) | 80 (1.8%) | 665 (14.9%) | 22.16 | 4,475 |
| **EDUCATION** | | | | | | | |
| NONE | 53 (10.0%) | 441 (82.9%) | 35 (6.6%) | 3 (0.6%) | 38 (7.2%) | 21.31 | 532 |
| PRIMARY | 661 (12.2%) | 4,253 (78.6%) | 444 (8.2%) | 50 (0.9%) | 494 (9.1%) | 21.44 | 5,408 |
| SECONDARY AND ABOVE | 309 (10.9%) | 2,185 (77.2%) | 292 (10.3%) | 44 (1.6%) | 336 (11.9%) | 21.69 | 2,827 |
| **RELIGION** | | | | | | | |
| MUSLIM | 122 (10.0%) | 948 (78.1%) | 124 (10.2%) | 20 (1.6%) | 144 (11.8%) | 21.86 | 1,214 |
| NON-MUSLIM | 901 (11.9%) | 5,928 (78.5%) | 647 (8.6%) | 77 (1.0%) | 724 (9.6%) | 21.46 | 7,553 |
| **CURRENT SMOKER** | | | | | | | |
| NO | 846 (11.0%) | 6,000 (78.3%) | 728 (9.5%) | 92 (1.2%) | 820 (10.7%) | 21.62 | 7,666 |
| YES | 177 (16.1%) | 876 (79.6%) | 43 (3.9%) | 5 (0.5%) | 48 (4.4%) | 20.79 | 1,101 |
| **HIV SEROSTATUS** | | | | | | | |
| NEGATIVE | 902 (11.9%) | 5,922 (78.1%) | 667 (8.8%) | 88 (1.2%) | 755 (10.0%) | 21.50 | 7,579 |
| POSITIVE | 121 (10.2%) | 954 (80.3%) | 104 (8.8%) | 9 (0.8%) | 113 (9.6%) | 21.61 | 1,188 |
| **CURRENT ART USE** | | | | | | | |
| NO | 20 (8.8%) | 187 (82.0%) | 21 (9.2%) | 0 (0.0%) | 21 (9.2%) | 21.56 | 228 |
| YES | 101 (10.5%) | 767 (79.9%) | 83 (8.6%) | 9 (0.9%) | 92 (9.5%) | 21.62 | 960 |
| **DRINKS ALCOHOL** | | | | | | | |
| NO | 535 (13.2%) | 3,163 (77.9%) | 324 (8.0%) | 39 (1.0%) | 363 (9.0%) | 21.35 | 4,061 |
| YES | 488 (10.4%) | 3,713 (78.9%) | 447 (9.5%) | 58 (1.2%) | 505 (10.7%) | 21.66 | 4,706 |
| **LAST ALCOHOL USE** | | | | | | | |
| 0 DAYS–1 WEEK | 345 (10.4%) | 2,602 (78.4%) | 329 (9.9%) | 44 (1.3%) | 373 (11.2%) | 21.72 | 3,320 |
| 1–4 WEEKS | 80 (9.7%) | 662 (80.4%) | 74 (9.0%) | 7 (0.9%) | 81 (9.9%) | 21.53 | 823 |

*(Continued)*

**Table 3.** (Continued)

| MALE PARTICIPANT CHARACTERISTICS | UNDERWEIGHT (%) | NORMAL (%) | OVERWEIGHT (%) | OBESE (%) | OVERWEIGHT/OBESE (%) | MEAN BMI | TOTAL |
|---|---|---|---|---|---|---|---|
| WITHIN THE LAST 12 MONTHS | 63 (11.2%) | 449 (79.8%) | 44 (7.8%) | 7 (1.2%) | 51 (9.0%) | 21.47 | 563 |
| MORE THAN 12 MONTHS | 535 (13.2%) | 3,163 (77.9%) | 324 (8.0%) | 39 (1.0%) | 363 (9.0%) | 21.35 | 4,061 |
| **TOTAL** | 1023 (11.7) | 6876 (78.4) | 771 (8.8) | 97 (1.1) | 868 (9.9%) | 21.52 | 8,767 |

Male BMI groups by demographic characteristics. HIV = Human Immunodeficiency Virus. ART = anti-retroviral therapy. Note that row percentages may not total to 100% due to rounding.

Bharmal et al. proposed that religious individuals often attend religious gatherings that involve celebratory foods high in fats and sugar, engage in religious media practices (such as watching prayers or sermons at home) that provide easy access to snacks, and may hold a perception of prayer as physical activity due to the changes in body position [35].

In this study, the probability of being overweight or obese increased with SES, similar to past research in Uganda [13, 36, 37], sub-Saharan Africa [3, 24, 25, 38] and low- and middle-income countries [39]. This trend differs from that of high-income countries where the prevalence of overweight and obesity decreases with wealth [40]. Changes in the lifestyle of those with more resources may contribute to being overweight or obese [7, 13]. However, it is thought that as sub-Saharan Africa undergoes the nutritional transition fueled by urbanization and globalization [41, 42], consumption of calorie-dense but nutrition-poor foods will increase the prevalence of obesity in lower SES groups as well [22].

The lower physical activity required for the occupations associated with obesity in this study (fishing, bar/restaurant work, and trading/shopkeeping) may play a role in increasing BMI. These occupations and related communities (fishing and trading) are in more urbanized regions of the Rakai District and individuals may have more exposure to the effects of urbanization, which has previously been reported as a risk factor for obesity. Several studies have found that urbanization is associated with increased weight [10], partly due to easy access to unhealthy foods [13, 25] and decreased physical activity [38].

For women only, alcohol use in the past year increased the probability of overweight/obesity while reporting alcohol use in the last week increased the probability of obesity. Studies on

**Table 4. Prevalence risk ratio of underweight, overweight, and obesity by participant sex.**

| BMI Category | Count (%) | Unadjusted PRR (95% CI) | Adjusted PRR (95% CI) | P-value |
|---|---|---|---|---|
| **Underweight** | | | | |
| Females (Ref.) | 682 (6.5) | - - | - - | - - |
| Males | 1,025 (11.7) | 1.80 [1.64, 1.98] | 1.75, [1.57, 1.97] | <0.001 |
| **Overweight** | | | | |
| Females (Ref.) | 2,216 (23.8) | - - | - - | - - |
| Males | 773 (8.8) | 0.28 [0.27, 0.30] | 0.24, [0.18, 0.34] | <0.001 |
| **Obesity** | | | | |
| Females (Ref.) | 1,026 (11.0) | - - | - - | - - |
| Males | 98 (1.1) | 0.10 [0.08, 0.12] | 0.14, [0.08, 0.24] | <0.001 |

PRR = prevalence risk ratio. CI = Confidence Interval. Ref. = reference group. Risk of overweight, obesity, or underweight for males and females. Adjusted for age group, community type, socioeconomic status, occupation, marital status, religion, smoking, current alcohol use, and/or last alcohol use. See S1 File for full analysis models, including inclusion of interaction terms.

**Table 5. Predictors of overweight/obesity from adjusted and unadjusted multivariable analysis models for male and female participants.**

| Male | | | | Female | | | |
|---|---|---|---|---|---|---|---|
| | Unadjusted PRR (95% CI) | Adjusted PRR (95% CI) | p-value | | Unadjusted PRR (95% CI) | Adjusted PRR (95% CI) | p-value |
| **Age group, years** | | | | **Age group, years** | | | |
| 15–19 (Ref.) | - - | - - | - - | 15–19 (Ref.) | - - | - - | - - |
| 20–29 | 4.13, [2.88, 5.93] | 2.93 [1.94, 4.43] | <0.001 | 20–29 | 3.01, [2.56, 3.53] | 1.78, [1.50, 2.09] | <0.001 |
| 30–39 | 7.64, [5.38, 10.85] | 4.72, [3.04, 7.32] | <0.001 | 30–39 | 5.06, [4.32, 5.92] | 2.38, [2.01, 2.81] | <0.001 |
| 40–49 | 7.32, [5.12, 10.45] | 4.84, [3.09, 7.58] | <0.001 | 40–49 | 4.87, [4.13, 5.74] | 2.41, [2.03, 2.86] | <0.001 |
| **Community type** | | | | **Community type** | | | |
| Agrarian (Ref.) | - - | - - | - - | Agrarian (Ref.) | - - | - - | - - |
| Fishing | 1.57, [1.35, 1.83] | 1.67, [1.40, 2.01] | <0.001 | Fishing | 1.60, [1.43, 1.80] | 1.25, [1.16, 1.35] | <0.001 |
| Trading | 1.21, [1.04, 1.41] | 1.09, [0.94, 1.27] | 0.260 | Trading | 1.39, [1.26, 1.53] | 1.17, [1.10, 1.25] | <0.001 |
| **SES** | | | | **SES** | | | |
| Lowest (Ref.) | - - | - - | - - | Lowest (Ref.) | | | |
| Low-middle | 1.72, [1.40, 2.10] | 1.71, [1.40, 2.09] | <0.001 | Low-middle | 1.48, [1.31, 1.67] | 1.34, [1.23, 1.46] | <0.001 |
| High-middle | 2.57, [2.13, 3.10] | 2.28, [1.89, 2.74] | <0.001 | High-middle | 1.96, [1.73, 2.21] | 1.57, [1.45, 1.71] | <0.001 |
| Highest | 2.26, [1.87, 2.73] | 2.51, [2.06, 3.06] | <0.001 | Highest | 1.94, [1.72, 2.18] | 1.72, [1.59, 1.87] | <0.001 |
| **Occupation** | | | | **Occupation** | | | |
| Agriculture/housework (Ref.) | - - | - - | - - | Agriculture/housework (Ref.) | - - | - - | - - |
| Bar/Restaurant | 2.82, [1.28, 6.23] | 2.21, [1.11, 4.41] | 0.025 | Bar/Restaurant | 1.84, [1.55, 2.17] | 1.29, [1.17, 1.42] | <0.001 |
| Fishing | 1.50, [1.22, 1.84] | 1.32, [1.03, 1.70] | 0.028 | Trade/Shopkeeper | 2.07, [1.85, 2.33] | 1.38, [1.29, 1.48] | <0.001 |
| Trade/Shopkeeper | 2.38, [1.97, 2.86] | 1.75, [1.45, 2.11] | <0.001 | Other | 0.80, [0.72, 0.88] | 1.06, [0.99, 1.15] | 0.102 |
| Other | 1.19, [1.01, 1.42] | 1.44, [1.21, 1.71] | <0.001 | | | | |
| **Marital status** | | | | **Marital status** | | | |
| Never married (Ref.) | - - | - - | - - | Never married (Ref.) | - - | - - | - - |
| Previously married | 2.15, [1.64, 2.94] | 1.35, [0.98, 1.86] | 0.063 | Previously married | 2.92, [2.53, 3.37] | 1.16, [1.01, 1.33] | 0.039 |
| Currently married | 4.14, [3.41, 5.04] | 1.97, [1.53, 2.54] | <0.001 | Currently married | 3.02, [2.68, 3.42] | 1.22, [1.07, 1.39] | 0.003 |
| **Education** | | | | **Education** | | | |
| None (Ref.) | - - | - - | - - | None (Ref.) | - - | - - | - - |
| Primary | 1.28, [0.93, 1.76] | 1.09, [0.79, 1.50] | 0.591 | Primary | 0.98, [0.83, 1.15] | | |
| Secondary/Tertiary | 1.66, [1.21, 2.30] | 1.31, [0.94, 1.82] | 0.105 | Secondary/Tertiary | 1.06, [0.89, 1.25] | | |
| **Religion** | | | | **Religion** | | | |
| Muslim (Ref.) | - - | - - | - - | Muslim (Ref.) | - - | - - | - - |
| Non-Muslim | 0.81, [0.68, 0.96] | 0.85, [0.71, 1.01] | 0.071 | Non-Muslim | 0.87, [0.77, 0.99] | 0.89, [0.82, 0.96] | 0.002 |
| **Current Smoker** | | | | **Current Smoker** | | | |
| No (Ref.) | - - | - - | - - | No (Ref.) | - - | - - | - - |
| Yes | 0.40, [0.31, 0.54] | 0.37, [0.27, 0.50] | <0.001 | Yes | 0.99, [0.76, 1.28] | | |
| **HIV serostatus** | | | | **HIV serostatus** | | | |
| Negative (Ref.) | - - | - - | - - | Negative (Ref.) | - - | - - | - - |
| Positive | 0.95, [0.79, 1.15] | | | Positive | 0.97, [0.87, 1.07] | | |
| **Current ART use** | | | | **Current ART use** | | | |
| Yes (Ref.) | - - | - - | - - | Yes (Ref.) | - - | - - | - - |
| No | 1.04, [0.66, 1.63] | | | No | 0.98, [0.75, 1.29] | | |
| **Drinks Alcohol** | | | | **Drinks Alcohol** | | | |
| No (Ref.) | - - | - - | - - | No (Ref.) | - - | - - | - - |
| Yes | 1.20, [1.06, 1.36] | 0.97, [0.74, 1.28] | 0.849 | Yes | 1.73, [1.59, 1.89] | 1.21, [1.10, 1.33] | <0.001 |
| **Last Alcoholic Drink** | | | | **Last Alcoholic Drink** | | | |

*(Continued)*

**Table 5.** (Continued)

| Male | | | | Female | | | |
|---|---|---|---|---|---|---|---|
| | Unadjusted PRR (95% CI) | Adjusted PRR (95% CI) | p-value | | Unadjusted PRR (95% CI) | Adjusted PRR (95% CI) | p-value |
| More than 12 months (Ref.) | - - | - - | - - | More than 12 months (Ref.) | - - | - - | - - |
| 1–4 weeks | 1.10, [0.88, 1.38] | 1.08, [0.79, 1.50] | 0.621 | 1–4 weeks | 1.53, [1.33, 1.76] | 0.94, [0.84, 1.06] | 0.293 |
| 0 days–1 week | 1.26, [1.10, 1.44] | 1.09, [0.83, 1.43] | 0.547 | 0 days–1 week | 1.94, [1.74, 2.17] | 1.02, [0.93, 1.13] | 0.632 |
| | | | | **Past Pregnancy** | | | |
| | | | | No (Ref.) | - - | - - | - - |
| | | | | Yes | 3.26, [2.86, 3.72] | 1.10, [0.93, 1.29] | 0.272 |

PRR = Prevalence risk ratio. CI = Confidence Interval. SES = socioeconomic status. HIV = Human Immunodeficiency Virus. ART = anti-retroviral therapy. Ref. = reference group. For males, HIV status and current ART use were not statistically significant on bi-variate analysis and were not included in the final analysis. For females, education, HIV status, current ART use, and smoking status were not statistically significant on bi-variate analysis and were not included in the final analysis. All other variables listed were adjusted for in the final model.

**Table 6. Predictors of obesity from adjusted and unadjusted multivariable analysis models for male and female participants.**

| Males | | | | Females | | | |
|---|---|---|---|---|---|---|---|
| Obese | Unadjusted PRR (95% CI) | Adjusted PRR (95% CI) | p-value | Obese | Unadjusted PRR (95% CI) | Adjusted PRR (95% CI) | p-value |
| **Age group, years** | | | | **Age group, years** | | | |
| 15–19 (Ref.) | - - | - - | - - | 15–19 (Ref.) | - - | - - | - - |
| 20–29 | 3.85, [0.86, 17.23] | 3.37, [0.70, 16.25] | 0.128 | 20–29 | 5.11, [3.47, 7.52] | 3.33, [2.13, 5.19] | <0.001 |
| 30–39 | 14.22, [3.43, 57.90] | 12.00, [2.19, 65.82] | 0.004 | 30–39 | 9.71, [6.66, 14.16] | 6.40, [4.07, 10.05] | <0.001 |
| 40–49 | 20.25, [4.90, 83.73] | 18.14, [3.19, 103.09] | <0.001 | 40–49 | 9.27, [6.33, 13.57] | 6.52, [4.13, 10.31] | <0.001 |
| **Community type** | | | | **Community type** | | | |
| Agrarian (Ref.) | - - | - - | - - | Agrarian (Ref.) | - - | - - | - - |
| Fishing | 1.46, [0.89, 2.39] | | | Fishing | 1.52, [1.31, 1.77] | 1.52, [1.28, 1.80] | <0.001 |
| Trading | 1.32, [0.82, 2.12] | | | Trading | 1.41, [1.23, 1.60] | 1.27, [1.11, 1.44] | <0.001 |
| **SES** | | | | **SES** | | | |
| Lowest (Ref.) | - - | - - | - - | Lowest (Ref.) | - - | - - | - - |
| Low-middle | 1.69, [0.82, 3.47] | 1.61, [0.78, 3.32] | 0.201 | Low-middle | 1.52, [1.26, 1.83] | 1.57, [1.30, 1.90] | <0.001 |
| High-middle | 3.75, [1.99, 7.04] | 3.23, [1.71, 6.10] | <0.001 | High-middle | 2.29, [1.93, 2.73] | 2.21, [1.85, 2.64] | <0.001 |
| Highest | 3.65, [1.96, 6.79] | 3.26, [1.68, 6.32] | <0.001 | Highest | 2.22, [1.86, 2.64] | 2.42, [2.01, 2.92] | <0.001 |
| **Occupation** | | | | **Occupation** | | | |
| Agriculture/housework (Ref.) | - - | - - | - - | Agriculture/housework (Ref.) | - - | - - | - - |
| Bar/Restaurant | 6.58, [0.84, 51.49] | 5.36, [0.83, 34.51] | 0.077 | Bar/Restaurant | 1.86, [1.53, 2.26] | 1.65, [1.34, 2.03] | <0.001 |
| Fishing | 1.33, [0.62, 2.84] | 2.00, [0.91, 4.37] | 0.083 | Trade/Shopkeeper | 2.15, [1.87, 2.46] | 1.84, [1.60, 2.11] | <0.001 |
| Trade/Shopkeeper | 3.49, [1.89, 6.46] | 2.99, [1.63, 5.65] | <0.001 | Other | 0.81, [0.69, 0.95] | 1.11, [0.94, 1.31] | 0.204 |
| Other | 1.72, [0.98, 3.03] | 2.50, [1.42, 4.38] | 0.001 | | | | |
| **Marital Status** | | | | **Marital Status** | | | |
| Never married (Ref.) | - - | - - | - - | Never Married (Ref.) | - - | - - | - - |
| Previously married | 2.93, [1.13, 7.60] | 1.17, [0.37, 3.69] | 0.791 | Previously married | 3.92, [3.04, 5.06] | 1.50, [1.10, 2.05] | 0.011 |
| Currently married | 7.03, [3.39, 14.55] | 1.69, [0.60, 4.76] | 0.317 | Currently married | 4.04, [3.19, 5.12] | 1.52, [1.14, 2.05] | 0.005 |
| **Education** | | | | **Education** | | | |
| None (Ref.) | - - | - - | - - | None (Ref.) | - - | - - | - - |
| Primary | 1.65, [0.51, 5.29] | | | Primary | 0.89, [0.72, 1.10] | | |

(Continued)

**Table 6.** (Continued)

| Males | | | | Females | | | |
|---|---|---|---|---|---|---|---|
| **Obese** | **Unadjusted PRR (95% CI)** | **Adjusted PRR (95% CI)** | **p-value** | **Obese** | **Unadjusted PRR (95% CI)** | **Adjusted PRR (95% CI)** | **p-value** |
| Secondary/Tertiary | 2.79, [0.86, 9.01] | | | Secondary/Tertiary | 1.00, [0.81, 1.25] | | |
| **Religion** | | | | **Religion** | | | |
| Muslim (Ref.) | - - | - - | - - | Muslim (Ref.) | - - | - - | - - |
| Non-Muslim | 0.61, [0.37, 1.01] | | | Non-Muslim | 0.84, [0.72, 0.98] | 0.80, [0.69, 0.94] | 0.005 |
| **Current Smoker** | | | | **Current Smoker** | | | |
| No (Ref.) | - - | - - | - - | No (Ref.) | - - | - - | - - |
| Yes | 0.38, [0.15, 0.93] | 0.35, [0.14, 0.88] | 0.026 | Yes | 1.02, [0.72, 1.45] | | |
| **HIV serostatus** | | | | **HIV serostatus** | | | |
| Negative (Ref.) | - - | - - | - - | Negative (Ref.) | - - | - - | - - |
| Positive | 0.65, [0.33, 1.29] | | | Positive | 0.83, [0.72, .97] | 0.59, [0.50, 0.69] | <0.001 |
| **Drinks Alcohol** | | | | **Drinks Alcohol** | | | |
| No (Ref.) | - - | - - | - - | No (Ref.) | - - | - - | - - |
| Yes | 1.29, [0.86, 1.94] | | | Yes | 1.64, [1.46, 1.84] | 1.09, [0.88, 1.36] | 0.431 |
| **Last Alcoholic Drink** | | | | **Last Alcoholic Drink** | | | |
| More than 12 months (Ref.) | - - | - - | - - | More than 12 months (Ref.) | - - | - - | - - |
| 1–4 weeks | 0.88, [0.39, 1.98] | | | 1–4 weeks | 1.40, [1.16, 1.68] | 1.03, [0.79, 1.33] | 0.840 |
| 0 days–1 week | 1.39, [0.90, 2.14] | | | 0 days–1 week | 1.95, [1.71, 2.23] | 1.34, [1.07, 1.68] | 0.012 |
| | | | | **Past Pregnancy** | | | |
| | | | | No (Ref.) | - - | - - | - - |
| | | | | Yes | 4.42, [3.38, 5.78] | 1.03, [0.72, 1.48] | 0.862 |

PRR = Prevalence risk ratio. CI = Confidence Interval. SES = socioeconomic status. HIV = Human Immunodeficiency Virus. Ref. = reference group. For males, education, religion, HIV status, current ART use, and last alcohol use were not statistically significant on bi-variate analysis and were not included in the final analysis. For females, education, smoking, current ART use, and smoking status were not statistically significant on bi-variate analysis and were not included in the final analysis. All other variables listed were adjusted for in the final model.

the association between drinking and BMI are conflicting, with many reporting little or no correlation between the two [43, 44]. However, several studies have suggested that binge or heavy drinking, rather than frequent light to moderate alcohol intake may be associated with increased BMI [45] and increased likelihood of being obese [46, 47]. The amount of alcohol intake was not available in this study.

Current smoking was inversely associated with being overweight/obese and was the only covariate that was positively associated with being underweight for males. Several studies in sub-Saharan Africa have similarly found smoking to be inversely associated with obesity in adults [13, 25, 48]. One study reported no association [11]. Nicotine has been shown to suppress appetite [49] and provide a behavioral alternative to eating [50], leading to decreased food intake and weight loss [50]. Smokers often gain weight after smoking cessation [50].

Importantly, HIV positive status was not associated with being underweight for males or females, although HIV positivity was inversely associated with obesity for females. HIV positivity was historically associated with lower BMI due to the health issues and the emaciated state of late-stage AIDS, however advances in HIV treatment have made the disease a manageable chronic disease. It has been proposed that stigma against the thinness associated with HIV may have promoted a cultural preference for a larger stature in African countries [26, 51]. This may no longer be the case, especially among younger generations who appear to value

**Table 7. Predictors of underweight from adjusted and unadjusted multivariable analysis models for male and female participants.**

| Males | Unadjusted PRR (95% CI) | Adjusted PRR (95% CI) | p-value | Females | Unadjusted PRR (95% CI) | Adjusted PRR (95% CI) | p-value |
|---|---|---|---|---|---|---|---|
| **Age group, years** | | | | **Age group, years** | | | |
| 15–19 (Ref.) | - - | - - | - - | 15–19 (Ref.) | - - | - - | - - |
| 20–29 | 0.28, [0.24, 0.33] | 0.35, [0.29, 0.42] | <0.001 | 20–29 | 0.51, [0.42, 0.63] | 0.97, [0.75, 1.26] | 0.837 |
| 30–39 | 0.34, [0.29, 0.39] | 0.45, [0.34, 0.58] | <0.001 | 30–39 | 0.35, [0.28, 0.44] | 0.73, [0.53, 1.02] | 0.062 |
| 40–49 | 0.53, [0.46, 0.62] | 0.64, [0.49, 0.83] | 0.001 | 40–49 | 0.55, [0.44, 0.68] | 1.06, [0.77, 1.45] | 0.739 |
| **Community type** | | | | **Community type** | | | |
| Agrarian (Ref.) | - - | - - | - - | Agrarian (Ref.) | - - | - - | - - |
| Fishing | 0.44, [0.36, 0.52] | 0.58, [0.46, 0.73] | <0.001 | Fishing | 0.56, [0.44, 0.71] | 0.66, [0.51, 0.86] | 0.002 |
| Trading | 0.92, [0.81, 1.04] | 0.99, [0.87, 1.12] | 0.868 | Trading | 0.76, [0.64, 0.91] | 0.82, [0.68, 0.97] | 0.023 |
| **SES** | | | | **SES** | | | |
| Lowest (Ref.) | - - | - - | - - | Lowest (Ref.) | - - | - - | - - |
| Low-middle | 0.90, [0.77, 1.04] | 0.86, [0.74, 0.99] | 0.039 | Low-middle | 0.70, [0.57, 0.85] | 0.68, [0.56, 0.84] | <0.001 |
| High-middle | 0.62, [0.52, 0.74] | 0.64, [0.54, 0.76] | <0.001 | High-middle | 0.57, [0.45, 0.71] | 0.56, [0.45, 0.71] | <0.001 |
| Highest | 0.78, [0.67, 0.91] | 0.62, [0.53, 0.73] | <0.001 | Highest | 0.66, [0.53, 0.81] | 0.57, [0.46, 0.71] | <0.001 |
| **Occupation** | | | | **Occupation** | | | |
| Agriculture/housework (Ref.) | - - | - - | - - | Agriculture/housework (Ref.) | - - | - - | - - |
| Bar/Restaurant | 1.32, [0.54, 3.25] | 1.46, [0.59, 3.62] | 0.418 | Bar/Restaurant | 0.76, [0.54, 1.08] | 0.92, [0.64, 1.34] | 0.683 |
| Fishing | 0.41, [0.31, 0.53] | 0.58, [0.42, 0.80] | 0.001 | Trade/Shopkeeper | 0.44, [0.33, 0.60] | 0.53, [0.39, 0.72] | <0.001 |
| Trade/Shopkeeper | 0.49, [0.39, 0.63] | 0.66, [0.52, 0.84] | 0.001 | Other | 1.25, [1.06, 1.47] | 0.90, [0.75, 1.09] | 0.283 |
| Other | 1.18, [1.04, 1.33] | 1.07, [0.94, 1.22] | 0.330 | | | | |
| **Marital status** | | | | **Marital status** | | | |
| Never married (Ref.) | - - | - - | - - | Never married (Ref.) | - - | - - | - - |
| Previously married | 0.70, [0.59, 0.83] | 0.95, [0.74, 1.22] | 0.683 | Previously married | 0.50, [0.41, 0.62] | 0.67, [0.47, 0.93] | 0.018 |
| Currently married | 0.47, [0.41, 0.53] | 0.72, [0.57, 0.91] | 0.006 | Currently married | 0.39, [0.33, 0.47] | 0.54, [0.43, 0.73] | <0.001 |
| **Education** | | | | **Education** | | | |
| None (Ref.) | - - | - - | | None (Ref.) | - - | - - | |
| Primary | 1.23, [0.94, 1.60] | | | Primary | 1.09, [0.81, 1.47] | | |
| Secondary/Tertiary | 1.10, [0.83, 1.45] | | | Secondary/Tertiary | 0.92, [0.67, 1.25] | | |
| **Religion** | | | | **Religion** | | | |
| Muslim (Ref.) | - - | - - | | Muslim (Ref.) | - - | - - | |
| Non-Muslim | 1.19, [0.99, 1.42] | | | Non-Muslim | 1.17, [0.92, 1.48] | | |
| **Current Smoker** | | | | **Current Smoker** | | | |
| No (Ref.) | - - | - - | - - | No (Ref.) | - - | - - | |
| Yes | 1.46, [1.25, 1.69] | 1.90, [1.60, 2.26] | <0.001 | Yes | 1.21, [0.79, 1.85] | | |
| **HIV status** | | | | **HIV status** | | | |
| Negative (Ref.) | - - | - - | | Negative (Ref.) | - - | - - | |
| Positive | 0.86, [0.72, 1.02] | | | Positive | 1.14, [0.95, 1.37] | | |
| **Currently on ART** | | | | **Currently on ART** | | | |
| Yes (Ref.) | - - | - - | | Yes (Ref.) | - - | - - | |
| No | 1.20, [0.76, 1.90] | | | No | 0.96, [0.61, 1.51] | | |
| **Drinks Alcohol** | | | | **Drinks Alcohol** | | | |
| No (Ref.) | - - | - - | - - | No (Ref.) | - - | - - | - - |
| Yes | 0.79, [0.70, 0.88] | 1.00, [0.78, 1.27] | 0.986 | Yes | 0.67, [0.56, 0.80] | 0.83, [0.59, 1.17] | 0.284 |
| **Last Alcoholic Drink** | | | | **Last Alcoholic Drink** | | | |

*(Continued)*

**Table 7.** (Continued)

| Males | | | | Females | | | |
|---|---|---|---|---|---|---|---|
| | **Unadjusted PRR (95% CI)** | **Adjusted PRR (95% CI)** | **p-value** | | **Unadjusted PRR (95% CI)** | **Adjusted PRR (95% CI)** | **p-value** |
| More than 12 months (Ref.) | - - | - - | - - | More than 12 months (Ref.) | - - | - - | - - |
| 1–4 weeks | 0.74, [0.59, 0.92] | 0.90, [0.66, 1.22] | 0.492 | 1–4 weeks | 0.78, [0.59, 1.02] | 1.18, [0.78, 1.78] | 0.436 |
| 0 days–1 week | 0.79, [0.69, 0.90] | 1.02, [0.79, 1.31] | 0.898 | 0 days–1 week | 0.60, [0.47, 0.77] | 0.987 [0.65, 1.44] | 0.868 |
| | | | | **Past Pregnancy** | | | |
| | | | | No (Ref.) | - - | - - | - - |
| | | | | Yes | 0.44, [0.37, 0.51] | 0.81, [0.58, 1.12] | 0.201 |

PRR = Prevalence risk ratio. CI = Confidence Interval. SES = socioeconomic status. HIV = Human Immunodeficiency Virus. ART = anti-retroviral therapy. Ref. = reference group. For males, education, religion, HIV status and current ART use were not statistically significant on bi-variate analysis and were not included in the final analysis. For females, education, religion, current smoking status, HIV status and current ART use were not statistically significant on bi-variate analysis and were not included in the final model. All other variables listed were adjusted for in the final model.

maintaining a healthy body weight [52, 53]. Additionally, weight gain and obesity is often associated with the initiation of ART in HIV positive individuals [54]. ART use was not associated with a higher probability of being overweight/obese in this population, which has high ART coverage (84%).

Past studies have reported higher education to be a risk factor for overweight/obesity in sub-Saharan Africa [23, 24, 29, 55], a trend that is opposite what is seen in high-income countries [56] and in South Africa [25]. It is proposed that educational attainment is associated with higher SES, leading to increased exposure to the risk factors for obesity. However, no significant association between education and the probability of being overweight or obese was found in this study.

## Limitations and strengths

This study has several limitations. First, BMI was used as a proxy for body fat, but is an imperfect assessment measurement due to differences in body composition and fat distribution. Secondly, there were limitations in the specificity of some covariates such as religion, occupation, and alcohol use. This prevented detailed characterization and analysis. Many variables, such as smoking or alcohol intake, were self-reported and may have been affected by social desirability bias or stigma. Additionally, the study has a primarily younger demographic, which limits its generalizability to older populations who may experience different associated factors. However, this study is one of the largest of its type to date, taking advantage of a large data set and conservative statistical analyses to provide valid findings with high power despite data limitations. Future studies could explore whether social and economic perceptions of body stature in rural Uganda affect the risk of obesity, what general awareness the population has of the risks of cardiovascular disease, and how awareness of this risk affects lifestyle choices.

## Conclusions

### Implications and recommendations

This study found the prevalence of overweight and obesity in non-urban communities of South Central Uganda to be 22.8% and 6.2%, respectively. Targeting healthy weight interventions to higher risk populations may be an effective way to effect change using interventions

optimized to fit the unique needs of specific community members [57, 58]. Therefore, there is an opportunity to increase awareness of the benefits of maintaining a healthy body weight amongst groups with higher prevalence of obesity identified in this study. As shifts in access to calorie-dense foods, increased wealth, and adoption of sedentary lifestyles occur, it is important to continue wide-spread education on the importance of healthy weight control, the benefits of healthy eating, and the cardiovascular risks of obesity.

## Supporting information

**S1 File.** Table A. Combined logistic regression model of overweight on covariates. Table B. Combined logistic regression model of obesity on covariates. Table C. Combined logistic regression model of underweight on covariates.
(DOCX)

## Acknowledgments

We thank all cohort participants, staff, and investigators of the Rakai Health Sciences Project. We thank Dr. Betsy Ogburn for her assistance with selecting an appropriate statistical method.

## Author Contributions

**Conceptualization:** Adeoluwa Ayoola, Aishat Mustapha, Anna Mia Ekstrom, Wendy S. Post, Larry W. Chang.

**Data curation:** Adeoluwa Ayoola, Robert Ssekubugu, Joseph Ssekasanvu, Wendy S. Post, Larry W. Chang.

**Formal analysis:** Adeoluwa Ayoola.

**Investigation:** Adeoluwa Ayoola, Mary Kathryn Grabowski, Joseph Ssekasanvu, Wendy S. Post, Larry W. Chang.

**Methodology:** Adeoluwa Ayoola.

**Project administration:** Joseph Ssekasanvu.

**Resources:** Joseph Ssekasanvu, Larry W. Chang.

**Software:** Adeoluwa Ayoola, Joseph Ssekasanvu.

**Supervision:** Wendy S. Post, Larry W. Chang.

**Visualization:** Adeoluwa Ayoola.

**Writing – original draft:** Adeoluwa Ayoola.

**Writing – review & editing:** Adeoluwa Ayoola, Robert Ssekubugu, Mary Kathryn Grabowski, Godfrey Kigozi, Aishat Mustapha, Steven J. Reynolds, Helena Nordenstedt, Rocio Enriquez, Ronald H. Gray, Maria J. Wawer, Joseph Kagaayi, Wendy S. Post, Larry W. Chang.

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
