## [Decision Letter · Decision Letter 0]

19 Jul 2022

PGPH-D-22-00872

Overweight and Obesity in South Central Uganda: A Population-Based Cohort Study

Dear Dr. Ayoola

Thank you for submitting your manuscript to PLOS Global Public Health. After careful consideration, we feel that it has merit but does not fully meet PLOS Global Public Health’s publication criteria as it currently stands. Therefore, we invite you to submit a revised version of the manuscript that addresses the points raised during the review process.

Based on the comments from the two reviewers, you manuscript still needs to be revised and corrected. Please attend to all the reviewers comments and make the necessary correction.

We look forward to receiving your revised manuscript.

Kind regards,

Zulkarnain Jaafar

Academic Editor

Journal Requirements:

1. Please provide a detailed online Financial Disclosure statement. This is published with the article. It must therefore be completed in full sentences and contain the exact wording you wish to be published.

a. Please clarify all sources of funding (financial or material support) for your study. List the grants (with grant number) or organizations (with url) that supported your study, including funding received from your institution. 

b. State the initials, alongside each funding source, of each author to receive each grant.

c. State what role the funders took in the study. If the funders had no role in your study, please state: “The funders had no role in study design, data collection and analysis, decision to publish, or preparation of the manuscript.”

d. If any authors received a salary from any of your funders, please state which authors and which funders.

Additional Editor Comments (if provided):

Reviewers' comments:

Reviewer's Responses to Questions

**Comments to the Author**

1. Does this manuscript meet PLOS Global Public Health’s publication criteria? Is the manuscript technically sound, and do the data support the conclusions? The manuscript must describe methodologically and ethically rigorous research with conclusions that are appropriately drawn based on the data presented.

Reviewer #1: Yes

Reviewer #2: Yes

2. Has the statistical analysis been performed appropriately and rigorously?

Reviewer #1: No

Reviewer #2: Yes

3. Have the authors made all data underlying the findings in their manuscript fully available (please refer to the Data Availability Statement at the start of the manuscript PDF file)?

Reviewer #1: Yes

Reviewer #2: Yes

4. Is the manuscript presented in an intelligible fashion and written in standard English?

Reviewer #1: Yes

Reviewer #2: Yes

5. Review Comments to the Author

Reviewer #1: Overall and interesting and well presented article. I have the following comments:

Title: I feel that the title is misleading since this is a cross-sectional study within a cohort study. Remove cohort from title.

Abstract: from the sentence starting with: "For female participants...." the Prevalence Risk Ratio (PRR) is used. However in the tables 5-7 the risk ratio (RR) is used. I am not sure which one is correct but expect it to be the RR.

Sample: Although it has been published elsewhere please provide a brief summary for the reader.

Line 87-89: provide the make and model of the anthropometer and scales and indicate how they were standardized.

Line 119: "To evaluate factors associated with underweight, overweight and obesity, we

used a modified Poisson model with a log link function to estimate prevalence risk ratios (PRR)." Was this the best method to use regarding the fact that the Poisson method is mainly associated with cohort studies not cross-sectional ones?

Tables: Please provide full titles for all tables so that they can stand alone.

It is not clear what < 12 months, < one month, < one week means. Rather write this out

It is recommended that tables 2 and 3 provide an additional column for BMI>=25 instead of only describing this in the text.

From Table 4 on please note the variables which have been adjusted for under each table.

Reviewer #2: Comments to authors

The authors assessed the prevalence of overweight/obesity, obesity and underweight and associated factor. Although the study lacks novelty, it will nonetheless contribute to literature on the subject as most Sub-Saharan counties lacks population-based studies on the subject.

In the methodology it was not clear which of the category of BMI was used as reference in the case of overweight/obesity and in the case of obesity as well as underweight. Because of this when the authors were describing the results the comparison did not quite come clearly.

I was thinking that because the dependent had three categories’ authors could have used multinomial logistic regression to estimate the risk of developing each condition.

Authors also used living in fishing community and fishing as different covariate to estimate the risk of overweight/obesity. I find these two variables as potential confounders which could mask, the effect of the other covariates as the probability of being a fisherman is high when one lives in a fishing community.

Authors may want to explain why these two variables were kept together in the model.

Also, authors were also silent about living in fishing community and fishing as a risk factor of high BMI in the discussion. Authors may want to provide some explanation on the association of these variables (living in fishing community and fishing). Perhaps it may have to do with how fishing is done in Uganda.

6. PLOS authors have the option to publish the peer review history of their article (what does this mean?). If published, this will include your full peer review and any attached files.

**Do you want your identity to be public for this peer review?** For information about this choice, including consent withdrawal, please see our Privacy Policy.

Reviewer #1: **Yes: **Prof Nelia Steyn

Reviewer #2: **Yes: **Abdulai Abubakari

---

## [Decision Letter · Decision Letter 1]

5 Oct 2022

Overweight and Obesity in South Central Uganda: A Population-Based Study

PGPH-D-22-00872R1

Dear Dr. Ayoola,

We are pleased to inform you that your manuscript 'Overweight and Obesity in South Central Uganda: A Population-Based Study' has been provisionally accepted for publication in PLOS Global Public Health.

Best regards,

Zulkarnain Jaafar

Academic Editor

Reviewer Comments (if any, and for reference):

Reviewer's Responses to Questions

**Comments to the Author**

1. If the authors have adequately addressed your comments raised in a previous round of review and you feel that this manuscript is now acceptable for publication, you may indicate that here to bypass the “Comments to the Author” section, enter your conflict of interest statement in the “Confidential to Editor” section, and submit your "Accept" recommendation.

Reviewer #2: All comments have been addressed

Reviewer #3: All comments have been addressed

2. Does this manuscript meet PLOS Global Public Health’s publication criteria? Is the manuscript technically sound, and do the data support the conclusions? The manuscript must describe methodologically and ethically rigorous research with conclusions that are appropriately drawn based on the data presented.

Reviewer #2: Yes

Reviewer #3: Yes

3. Has the statistical analysis been performed appropriately and rigorously?

Reviewer #2: Yes

Reviewer #3: I don't know

4. Have the authors made all data underlying the findings in their manuscript fully available (please refer to the Data Availability Statement at the start of the manuscript PDF file)?

Reviewer #2: Yes

Reviewer #3: Yes

5. Is the manuscript presented in an intelligible fashion and written in standard English?

Reviewer #2: Yes

Reviewer #3: Yes

6. Review Comments to the Author

Reviewer #2: (No Response)

Reviewer #3: Dear PLOS Global Public Health,

I recommend this article to be accepted. Please find below my main comments on this article.

- The title is reflecting the study

- The rationale and purpose of this study is clear.

- In the methodology, the recruitment, data collection methods, and plan of analysis is clearly explained

- Ethical consideration is taken and mentioned clearly.

- Findings/results are stated in a direct way with supportive tables

- Discussion is good and reflecting the objective of this research

- Limitation is stated with an emphasis on the strength of this study

- Conclusion is reflecting the main results and including recommendations to improve practice

7. PLOS authors have the option to publish the peer review history of their article (what does this mean?). If published, this will include your full peer review and any attached files.

**Do you want your identity to be public for this peer review?** For information about this choice, including consent withdrawal, please see our Privacy Policy.

Reviewer #2: **Yes: **Prof. Abdulai Abubakari

Reviewer #3: No
